

# Comparison of kinematics and electromyography in the last repetition during different maximum repetition sets in the barbell back squat

Hallvard Nygaard Falch, Andreas Hegdahl Gundersen, Stian Larsen, Markus Estifanos Haugen and Roland van den Tillaar

Department for Sports Science and Physical Education, Nord University, Levanger, Trøndelag, Norway

## ABSTRACT

**Background:** The barbell squat is an exercise used to strengthen the lower limbs, with implications for both performance in sports and improving movement during everyday tasks. Although the exercise is being trained across a variety of repetition ranges, the technical requirements may vary, affecting appropriate repetition range for specific training goals.

**Methods:** A randomised within-subject design was used to compare kinematics and surface electromyography (EMG) in the lower extremities during different concentric phases (pre-, sticking- and post-sticking region) of the last repetition when performing squats at different repetition maximums (RMs). Thirteen strength-trained men (age: 23.6 ± 1.9 years; height: 181.1 ± 6.5 cm; body mass: 82.2 kg, 1RM: 122.8 ± 16.2, relative strength: 1.5 ± 0.2 x body mass in external load) performed a 1, 3, 6, and 10RM squat, in a randomised order.

**Results:** The main findings were that barbell-, ankle-, knee- and hip kinematics were similar across different repetition ranges, except for a smaller trunk lean at 1RM in the pre-sticking region compared to other repetitions and in the sticking region compared to 10RM ($p \leq 0.04$). Furthermore, 1RM revealed significantly higher EMG amplitude in the vastus lateralis, gastrocnemius and soleus in the sticking and post-sticking regions when compared to 10RM. It was concluded that 10RM may locally fatigue the vastus lateralis and plantar flexors, explaining the lower EMG amplitude. The observed differences indicate that requirements vary for completing the final repetition of the 10RM compared to the 1RM, an important aspect to consider in training to enhance 1RM strength.

## INTRODUCTION

Resistance training is a common training modality for improving health in the general public, as it may improve basal metabolic rate, decrease blood pressure, improve insulin sensitivity, and reduce muscle loss in older adults (*Kraemer, Ratamess & French, 2002*), while also being used as a tool for improving sports performance among athletes

Corresponding author
Hallvard Nygaard Falch,
falchuci@gmail.com

(*Falch, Rædergård & van den Tillaar, 2019*; *Rædergård, Falch & van den Tillaar, 2020*). A common exercise utilised is the bilateral back squat, which is often incorporated into training programmes for enhancing lower-body strength (*Comfort, McMahon & Suchomel, 2018*). Furthermore, the barbell back squat is a competition lift in powerlifting, where the technical execution varies, as athletes attempt to optimize factors like bar placement and stance according to their individual anthropometrics. For an attempt to be considered valid, the athlete must descend until the hip joint is vertically lower than the knee joint. After reaching the approved depth, the athlete has to raise the barbell by extending the hip and knees until an initially erect position to complete the lift. Performance in powerlifting is determined by maximal weight lifted for one repetition maximum (1RM) (*Kraemer & Ratamess, 2004*). The 1RM is a display of maximal strength, as strength is often defined as maximum force produced against external resistance (*Stone, Stone & Lamont, 1993*). In this context, maximal strength is task-specific, as different strength tasks require unique technical adaptations. Athletes must optimize their movement according to their individual anthropometrics and capacities, such as motor unit recruitment, firing rate, limb lengths, muscle mass, and joint architecture (*Suchomel et al., 2018*; *Lovera & Keogh, 2015*).

For improving strength, it has traditionally been accepted that adaptions from a set of resistance training may be influenced by the repetition range (*Schoenfeld et al., 2015*), a concept known as the "repetition continuum" (*Schoenfeld et al., 2021*). Thus, the relative load chosen for a set, which is negatively proportional to the number of repetitions, is a decision made on the specific training goal (hypertrophy, maximal strength, strength endurance) (*Shimano et al., 2006*). The repetition continuum has been questioned in more recent research as hypertrophy can be achieved to a similar degree with varying loads, while heavy loads are still deemed superior for increasing strength (*Schoenfeld et al., 2021*). Training with heavier loads is suggested to increase force-producing capabilities by enhancing neuromuscular function (*Haff & Triplett, 2015*). For example, *Jenkins et al. (2017)* observed increased isometric electromyographic (EMG) amplitude of the quadriceps after leg extension training with 80% of 1RM, while training at 30% of 1RM and did not significantly increase EMG amplitude. It was concluded that training with 80% of 1RM resulted in greater increases in strength, despite a similar increase in muscle thickness, which was speculated to be attributed to neurological factors.

Despite increasing acceptance for similar hypertrophy at low *vs* high load training (*Schoenfeld et al., 2021*), the practical applicability of the "repetition continuum" and how it applies to the 1RM squat is not fully understood. Although "heavy" loads are favourable for increasing strength, there is uncertainty regarding the threshold for a load to be deemed specific to the 1RM squat, as the specificity of load is attributed to research studying varying exercises and repetition ranges (*Schoenfeld et al., 2021*). Although 1RM training is more specific to the competition lift, training with other repetition ranges (*e.g.*, 3RM) undoubtedly has benefits, such as allowing more repetitions (*Nuzzo et al., 2023*). Thus, practice at a high volume with sets consisting of higher repetition ranges could be of great importance when considering long-term increases in strength (*Kraemer et al., 2002*).
However, many sets are performed until exhaustion in which fatigue occurs. *Brice et al. (2020)* observed fatigue-induced compensatory strategies when approaching failure in squatting. They suggested that knee-joint loading decreases, while hip-joint moment increases as failure approaches. However, it is yet to be investigated if compensatory strategies are load dependent and/or vary for the different phases when ascending in the squat. A percentage of 1RM is commonly used to prescribe training loads. However, it is unclear if technique remains consistent across the entire spectrum of these loads when training to, or near voluntary failure. Earlier research supports the assumption that technical variations depend on load, as evidenced by the influence of load on EMG signals (*Mehls et al., 2021*; *Martinez, Coons & Mehls, 2023*). Repetition range might influence joint and barbell kinematics and EMG amplitude during different phases of the lift due to compensatory strategies, which are indirect measures (*Morton et al., 2019*; *Larsen, Haugen & van den Tillaar, 2022*) of understanding differences between a given repetition range and 1RM squat strength. When attempting to complete the lift, athletes experience several events in the ascending phase, including the sticking region, as a result of a mechanical disadvantage that the athlete has to grind through to complete the lift (*Suchomel et al., 2018*). Athletes usually fail a lift within the sticking region (*van den Tillaar, Andersen & Saeterbakken, 2014*), whereby the occurrence of the sticking region and compensatory strategies is indicated to be linked with effort and fatigue (*Newton et al., 1997*; *Brice et al., 2020*).

Although several mechanisms can lead to concentric neuromuscular failure (*Larsen, Haugen & van den Tillaar, 2022*), similar EMG and kinematics in the different phases of the last repetition when performing squats of different repetition ranges could indicate similar requirements to grind through the sticking region. This knowledge could aid in constructing training programs tailored to specific goals by selecting a repetition range with similar technical requirements.

An investigation conducted by *Larsen et al. (2022)* in the bench press, involving 1, 3, 6, and 10RM, observed similar EMG and kinematics in the final repetition, with the exception of a higher velocity in the 10RMs final repetition compared to the 1RM. However, it is unknown whether these findings also apply to the barbell back squat, as training the lower body is suggested to be more susceptible to non-local muscular fatigue (*Halperin, Chapman & Behm, 2015*).

The objective of this research was to compare kinematics and EMG amplitude in the lower extremities during the concentric phase of the last repetition when performing the bilateral back squat at different RMs (1, 3, 6, and 10RM). Based on the findings from *Larsen et al. (2022)*, it was expected that barbell velocity would be higher for 10RM compared to the other repetition ranges due to greater barbell inertia, with no significant difference in EMG amplitude and joint kinematics, since a maximal voluntary concentric contraction is required to complete the final repetition (*Morton et al., 2016*).

## MATERIALS AND METHODS

A randomized within-subject design was utilized to compare kinematics and EMG amplitude in the lower body during the last repetition of the bilateral back squat at
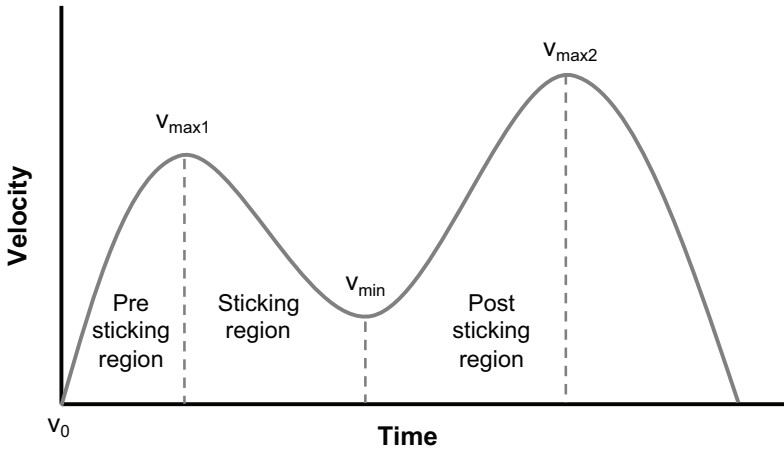

**Figure 1 The different phases of the sticking region.**

different RMs. This study investigated the effect of repetition number (1, 3, 6, and 10RM) on EMG amplitude and kinematics across various phases of the last repetition in the parallel back squat and phases (pre-sticking, sticking, and post-sticking region). The different events when ascending are as follows: $V_0$: bottom position; $V_{max1}$: first peak velocity; $V_{min}$: first minimum velocity; and $V_{max2}$: second peak velocity. These events divide the different phases (Fig. 1) into the pre-sticking ($V_0$–$V_{max1}$), sticking ($V_{max1}$–$V_{min}$), and post-sticking regions ($V_{min}$–$V_{max2}$) when ascending in the squat (*van den Tillaar, Andersen & Saeterbakken, 2014*, *2019*) (Fig. 1).

An *a priori* power analysis was calculated to determine adequate sample size using G*Power (version 3.1.9.2; University of Kiel, Kiel, Germany). Using effect size $f$ of 0.8, $\alpha = 0.05$ and based upon the findings of *Larsen et al. (2022)* a total sample of $n = 12$ would be sufficient to find significant and medium-sized effects with an actual power of 0.80. Thirteen recreationally strength-trained men (age: 23.6 ± 1.9 years; height: 181.1 ± 6.5 cm; body mass: 82.2 kg, relative strength: 1.5 ± 0.2 x body mass in external load) participated in the study. The subjects were required to back squat a minimum of 1.2 x body mass with a technique fulfilling the criteria set by the International Powerlifting Federation (*Spence et al., 2022*). Each subject had to declare absence of injury or any other illness which could negatively affect performance. Furthermore, subjects were instructed to meet in a prepared state (*e.g.*, consumption of a light meal, enough sleep, withdrawal of alcohol and heavy resistance training 48 h prior to testing) without consumption of caffeine on the given day. The methodology, risks, and benefits were explained both orally and in text, whereby written consent had to be signed prior to participation. The study was approved by the local ethics committee (Project number: 701688) and conformed to the latest revision of the Helsinki Declaration.

Each subject performed two sessions, separated by at least 72 h of rest: the first session was required for familiarisation with the protocol and to establish the load for the different repetition ranges, while data collection was conducted in the second session. For each attempt within both sessions, the stance width was standardised based upon the subjects' own personal preference to increase ecological validity. However, lifting equipment

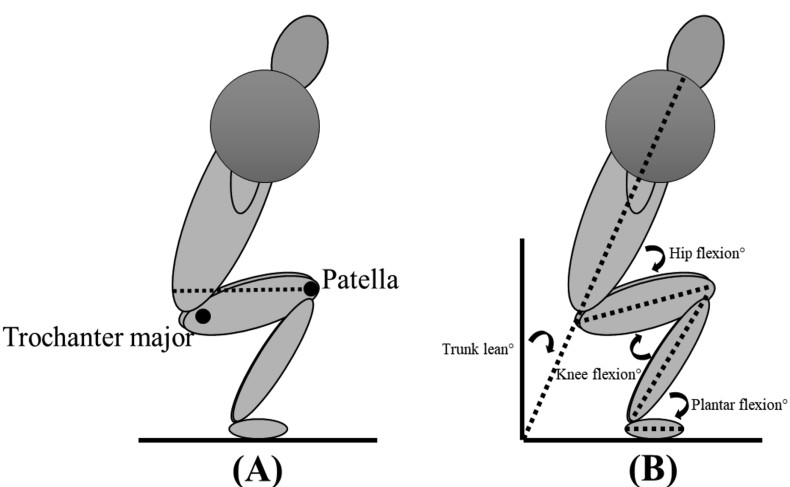

**Figure 2 Depth requirements for the bilateral back squat (A) and definition of joint angles (B).**

(*e.g.*, knee sleeves, lifting belt) was not allowed, with the exception of lifting shoes. The subjects were not restricted in concentric/eccentric tempo, but were not allowed to remain at lockout for longer than 2 s. The sessions were performed in a similar manner, starting with a standardised specific warm-up performing the squat at different percentages (40%, 60%, 70% and 80%) of estimated 1RM (*Gomo & van den Tillaar, 2016*), before performing a 1, 3, 6-, and 10RM back squat in a randomised order determined by an online randomiser (https://www.random.org/lists/). Preferred stance width and body height were measured in the familiarisation session with measuring tape, while body mass was measured with a standing scale (Soehnle professional 7830, stand scale). Appropriate depth was defined according to the International Powerlifting Federation, requiring the trochanter major to be vertically lower than the patella (Fig. 2A), which was controlled for by a 3D motion capture system. The subject had to rest a minimum of 5 min between each attempt to reduce the chance of fatigue influencing performance outcomes.

A 3D motion capture system (Qualisys, Gothenburg, Sweden) with eight cameras sampling at 500 Hz tracking reflective markers was used to determine sagittal-plane joint kinematics of the hip, knee, and ankle joints defined as 0° in a standing upright position (Fig. 2B). Reflective markers were placed on anatomical landmarks (acromion, pelvis, iliac crest, posterior superior iliac spine, trochanter major, medial and lateral condyle of the knee, medial and lateral malleolus, sternum, tuber calcanei, 1st and 5th proximal phalanx). Trunk lean was defined relative to the laboratory. Kinematic data collected by the 3D system were exported to Visual 3D (C-Motion, Inc., Washington, DC, USA) as C3D files for further analysis.

EMG was recorded with the use of Musclelab 6000 (Ergotest Technology AS, Langesund, Norway). EMG electrodes (Zynex Neurodiagnostics, Lone Tree, CO, USA) were placed on 10 muscles (erector spinae iliocostalis, medium and maximus gluteus, rectus femoris, vastus medialis and lateralis, biceps femoris, semitendinosus, gastrocnemius, and soleus) of the subjects' self-percieved dominant side based upon

strength, after appropriate preparation (shaving, rasping, and cleaning with alcohol) according to the recommendations of *Hermens et al. (2000)*. The electrodes (11 mm contact diameter and a 2 cm centre-to-centre distance) were placed along the presumed direction of the underlying muscle fibre. EMG signals were sampled at a rate of 1,000 Hz, with a preamplifier utilised to increase the signal-to-noise ratio. Root mean square (RMS) of the raw EMG signal was calculated by a hardware circuit network (frequency response 20–500 Hz, integrating moving average filter with 100 ms width, total error ± 0.5%). None-normalized EMG can be reliably used for comparisons within a single session and in unchanged settings (*Halaki & Ginn, 2012*). As the current study employed a within-subject design, whereby the participants performed squats with standardized requirements, within one session, without inter-muscular comparison. Thus, eliminating the need for EMG normalization, while also reducing test duration.

Mean RMS EMG signals during the different phases of the lift were found by synchronising EMG with a linear encoder (ET-Enc-02; Ergotest Technology AS, Langesund, Norway). The linear encoder was attached to the barbell, which measured with a resolution of 0.019 mm and a sampling rate of 200 Hz. Data from the EMG and the linear encoder were synchronised using a data synchronisation unit and analysed in Musclelab v.10.200.90.5095 (Ergotest Technology AS, Langesund, Norway). Both EMG and kinematic data were sampled from the different events of the last repetition of the ascending phase and used for statistical analysis.

## Statistical analysis

To compare joint kinematics and barbell velocity across different repetition ranges (1–10RM), a one-way analysis of variance (ANOVA) with repeated measures at the different events ($V_0$, $V_{max1}$, $V_{min}$, and $V_{max2}$) was assessed. A 4 (repetition range: 1, 3, 6, and 10RM) by 3 (phase: pre-sticking, sticking, and post-sticking regions) with repeated measures was assessed to compare EMG amplitude for each muscle through the different repetition ranges across the different phases. When significant differences were observed, the *p*-value was corrected for *post hoc* by the Holm–Bonferroni correction. The assumption of sphericity was controlled for by Mauchly's test of sphericity. If the assumption of sphericity was violated, the Greenhouse–Geisser adjusted *p*-value was reported. The level of significance was set at $p < 0.05$. Data are reported as means ± standard deviations. Effect size was evaluated as eta partial squared ($\eta_p^2$), whereby 0.01 to 0.06 $\eta_p^2$ was defined as a small effect, 0.06 to 0.14 $\eta_p^2$ a medium effect, and 0.14 > $\eta_p^2$ a large effect (*Cohen, 1988*). The statistical analysis was conducted in IBM SPSS Statistics 27.0 (IBM, Armonk, NY, USA).

## RESULTS

Thirteen subjects met the inclusion criteria (Table 1).

Only significant differences were observed for trunk lean at $V_{max1}$ and $V_0$, and knee flexion at $V_{min}$ (F ≥ 3.34, $p \leq 0.05$, $\eta_p^2 \geq 0.25$). *Post-hoc* tests revealed that trunk lean was significantly lower at $V_0$ for 1RM compared to all other loading ranges and at $V_{max1}$ compared to 10RM ($p \geq 0.04$). Furthermore, knee flexion was significantly lower at 6RM compared to 1RM at $V_{min}$ ($p = 0.01$). No significant difference was observed for hip angle

**Table 1 Descriptive statistics of the participants presented in mean ± standard deviation.**

|  | Mean ± Standard deviation |
| --- | --- |
| 1-RM (kg) | 122.8 ± 16.2 |
| 3-RM (kg) | 111.4 ± 14.4 |
| 6-RM (kg) | 101.3 ± 15 |
| 10-RM (kg) | 92.3 ± 13.8 |
| Relative strength (1-RM/body mass) | 1.5 ± 0.2 |

**Table 2 Descriptive statistics of knee-, hip-, ankle- and trunk angle and barbell velocity at the different events of the last repetition during 1-, 3-, 6-, and 1-RM barbell back squat.**

|  | 1-RM | 3-RM | 6-RM | 10-RM |
| --- | --- | --- | --- | --- |
| **$V_0$** | | | | |
| Knee flexion (°) | 124.4 ± 8.2 | 125.1 ± 8.8 | 124.8 ± 8.6 | 125.4 ± 9.7 |
| Hip flexion (°) | 96.7 ± 12.3 | 97.4 ± 13.1 | 95.3 ± 15.0 | 96.7 ± 13.1 |
| Ankle dorsal flexion (°) | 21.1 ± 5.7 | 18.2 ± 5.8 | 18.5 ± 6.7 | 18.2 ± 7.7 |
| Trunk lean (°) | 50.5 ± 6.2[*] | 52.5 ± 6.1 | 52.4 ± 7.0 | 53.9 ± 6.9 |
| **$V_{max1}$** | | | | |
| Knee flexion (°) | 113.2 ± 10 | 112.7 ± 8.3 | 112.1 ± 8.3 | 113.5 ± 9.8 |
| Hip flexion (°) | 90.9 ± 10.5 | 90.2 ± 9.8 | 88.2 ± 14.2 | 90.1 ± 10.6 |
| Ankle dorsal flexion (°) | 18.1 ± 6.5 | 15.2 ± 5.8 | 15.2 ± 6.8 | 15.2 ± 7.7 |
| Trunk lean (°) | 54.2 ± 6.1 | 55.9 ± 6.0 | 56.0 ± 6.6 | 57.0 ± 6.1[†] |
| Barbell velocity (m/s) | 0.28 ± 0.05 | 0.32 ± 0.07 | 0.33 ± 0.08 | 0.33 ± 0.08 |
| **$V_{min}$** | | | | |
| Knee flexion (°) | 85.9 ± 12.2 | 84.1 ± 11.6 | 79.9 ± 7.8[†] | 82.5 ± 9.3 |
| Hip flexion (°) | 72.4 ± 10.2 | 71.5 ± 6.4 | 67.2 ± 10.7 | 70.0 ± 8.4 |
| Ankle dorsal flexion (°) | 11.6 ± 6.4 | 7.2 ± 6.4 | 6.3 ± 6.7 | 7.2 ± 7.7 |
| Trunk lean (°) | 56.1 ± 6.2 | 58.6 ± 7.1 | 58 ± 7.4 | 59.0 ± 7.8 |
| Barbell velocity (m/s) | 0.08 ± 0.05 | 0.09 ± 0.07 | 0.11 ± 0.05 | 0.12 ± 0.05 |
| **$V_{max2}$** | | | | |
| Knee flexion (°) | 43.2 ± 8.6 | 42.7 ± 6.6 | 43.1 ± 7.1 | 42.9 ± 8.7 |
| Hip flexion (°) | 34.5 ± 14.0 | 33.1 ± 16.2 | 32.5 ± 15.5 | 33.9 ± 13.4 |
| Ankle dorsal flexion (°) | 2.4 ± 6.4 | 0.0 ± 4.8 | 0.1 ± 7.0 | 0.6 ± 8.0 |
| Trunk lean (°) | 39.5 ± 6.4 | 40.4 ± 7.4 | 40.2 ± 7.6 | 41.5 ± 6.8 |
| Barbell velocity (m/s) | 0.55 ± 0.16 | 0.6 ± 0.13 | 0.57 ± 0.13 | 0.52 ± 0.14 |

Notes:
[*] Indicates a significant different joint angle compared to all other loads at a $p < 0.05$ level.
[†] Indicates a significant different joint angle compared to when performing a 1-RM.

and barbell velocity when comparing the different repetition ranges (F ≤ 1.61, $p \geq 0.21$, $\eta_p^2 \leq 0.15$) (Table 2).

A significant effect of repetition range was observed only for the vastus lasteralis, soleus, and gastrocnemius at the sticking region and post-sticking region and for the gastrocnemius at the pre-sticking region (F ≥ 3.35, $p \leq 0.04$, $\eta_p^2 \geq 0.32$), while no significant effect of repetition range was observed for any of the other muscles (F ≤ 1.84, $p \leq 0.17$,

**Table 3 Mean EMG (RMS) in the pre-, sticking-, and post- sticking region for the different muscles across different repetition ranges.**

| | Semitendinosus | Erector spinae | Rectus femoris | Vastus medialis | Vastus lateralis | Gluteus medius | Gluteus maximus | Soleus | Gastrocnemius | Biceps femoris |
|---|---|---|---|---|---|---|---|---|---|---|
| **Pre-sticking region** | | | | | | | | | | |
| 1-RM | 46.9 ± 34.1 | 260.3 ± 119.7 | 434.9 ± 228 | 343.2 ± 233.6 | 379.9 ± 219 | 40 ± 33.2[*†] | 54.8 ± 54.9[*†] | 212.9 ± 105.5[†] | 104.8 ± 46[*†] | 99.2 ± 69 |
| 3-RM | 74.6 ± 21.2[†] | 286.1 ± 105[†] | 494.4 ± 234.4 | 235.4 ± 171.3 | 379.7 ± 221.8 | 45.4 ± 27.5[*†] | 59.4 ± 45.4[*†] | 184.5 ± 93.7 | 83.4 ± 28.3[†] | 103.2 ± 69 |
| 6-RM | 32.6 ± 22.5[†] | 302.3 ± 84.8[†] | 479.8 ± 269.4 | 307.3 ± 275.5[†] | 342.4 ± 218.8 | 39.2 ± 29.7[*†] | 48.7 ± 38.6[*†] | 190.8 ± 77.1[*†] | 111.7 ± 77.1[*†] | 80.5 ± 54.5 |
| 10-RM | 45.6 ± 47.1 | 245.1 ± 81.8 | 429.1 ± 283.8 | 277.9 ± 214.7[†] | 327.2 ± 202.4 | 44.7 ± 33.2[*†] | 52.2 ± 51.3[*†] | 133.8 ± 78.4 | 68.9 ± 34.4[†] | 76.8 ± 58.9 |
| **Sticking region** | | | | | | | | | | |
| 1-RM | 66.1 ± 41.8 | 259.2 ± 105.9 | 463 ± 251.9 | 330.1 ± 242.9 | 369.5 ± 185.1[†] | 81.5 ± 51.7[†] | 102.7 ± 63.7 | 164.5 ± 57.8 | 60.9 ± 18.5 | 116.9 ± 63.7 |
| 3-RM | 69.6 ± 35.5 | 268.6 ± 88.3 | 449.8 ± 229.2 | 241.5 ± 192 | 404.4 ± 230.7[†] | 87.4 ± 47.1[†] | 119.8 ± 65.5 | 151.3 ± 66.6 | 66.8 ± 39.3 | 127.3 ± 63.3 |
| 6-RM | 69 ± 40.9[†] | 268.9 ± 80.8[†] | 427.6 ± 205.7 | 287.1 ± 236.1[†] | 349.2 ± 196.4[†] | 78.9 ± 45.3[†] | 120 ± 69.9 | 136 ± 59.7 | 64.4 ± 25[†] | 128.8 ± 63.9 |
| 10-RM | 70.7 ± 43.2[†] | 255.3 ± 97.3[†] | 401.2 ± 204.8 | 269.3 ± 198.1[†] | 311.5 ± 159.4[†] | 79.5 ± 42.3[†] | 93.5 ± 55[†] | 125.4 ± 46.9[†] | 57 ± 40.7 | 107.8 ± 60.9 |
| **Post-sticking region** | | | | | | | | | | |
| 1-RM | 76 ± 50.1 | 248.7 ± 95.5 | 318.8 ± 152.3 | 288 ± 218.3 | 334.3 ± 163.8 | 125.8 ± 68.6 | 118.2 ± 69.6 | 143.9 ± 57.9 | 57.2 ± 28.8 | 151.9 ± 69.6 |
| 3-RM | 84.2 ± 49.5 | 225.9 ± 81.1 | 308.8 ± 148 | 210.8 ± 117.6 | 326.1 ± 160.6 | 135.9 ± 80.4 | 128.6 ± 68.8 | 139.7 ± 64.3 | 53.2 ± 26.2 | 148.7 ± 77.7 |
| 6-RM | 78.2 ± 51.4 | 237.7 ± 81 | 292.8 ± 133.3 | 225.2 ± 187.2 | 303.9 ± 151.1 | 125.1 ± 63.1 | 124.3 ± 67.3 | 117.6 ± 46.1 | 45.1 ± 25.7 | 132.5 ± 51.8 |
| 10-RM | 69.2 ± 48.6 | 211.6 ± 77.5 | 254.7 ± 125.9 | 221.2 ± 200.3 | 265.3 ± 135.2 | 106.3 ± 49.8 | 104.5 ± 63.8 | 102.4 ± 45.7 | 39.3 ± 15.7 | 134.5 ± 55.8 |

**Notes:**
[*] Indicates a significant difference to the sticking region on a $p < 0.05$ level.
[†] Indicates a significant difference to the post-sticking region on a $p < 0.05$ level.

$\eta_p^2 \leq 0.21$). A significant effect of the different phases was observed for all muscles (F ≥ 3.62, $p \leq 0.05$, $\eta_p^2 \geq 0.27$), except for the biceps femoris and rectus femoris (F ≤ 4.53, $p \leq 0.06$, $\eta_p^2 \leq 0.39$). No significant interaction effect was observed for EMG amplitude across the different repetition ranges and phases of the lift (F ≤ 2.11, $p \leq 0.08$, $\eta_p^2 \leq 0.21$) (Table 3).

*Post-hoc* analysis revealed significantly higher EMG amplitude in the vastus lateralis in the sticking region and post-sticking region when performing 1RM compared to 10RM ($p \leq 0.03$) and in the post-sticking region for 3RM compared to 10RM. Soleus EMG amplitude was significantly higher at 1RM compared to 6 and 10RM in both the sticking and post-sticking regions ($p \leq 0.02$), while gastrocnemius EMG amplitude was significantly higher at 1RM compared to 10RM in the pre-, sticking, and post-sticking regions and when compared to 6RM in the post-sticking region ($p \leq 0.02$). EMG amplitude was also significantly higher for the soleus at 3RM compared to 10RM in the pre- and post-sticking regions (Fig. 3).

## DISCUSSION

The objective of the current study was to compare kinematics and EMG amplitude in the lower body in the last repetition when performing the barbell back squat of different RM (1, 3, 6, and 10RM). The main findings were that barbell velocity was not influenced by repetition range, in contrast to a similar analysis in bench press (*Larsen, Haugen & van den Tillaar, 2022*) that observed a higher velocity at 10RM in the sticking region compared to 1 and 3RM. However, in comparison to 1, 10RM revealed more trunk lean at $V_{max1}$ and $V_0$, accompanied by lower EMG amplitude for the vastus lateralis and soleus in the sticking and post-sticking regions and in the gastrocnemius for all phases of the lift.

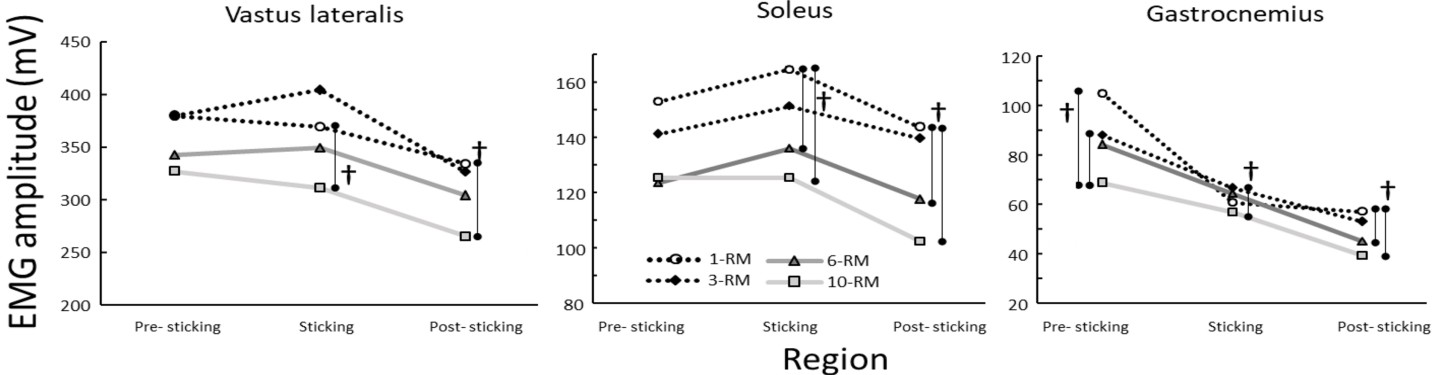

**Figure 3 Mean EMG amplitude for the vastus lateralis, soleus and gastrocnemius through the different phases when performing the 1-, 3-, 6- and 10-RM barbell back squat.** [†]Indicates a significant different joint angle between these two repetition ranges at a $p < 0.05$ level.

According to the findings of the current study, the observed velocity during the squat at the different events suggests that the final repetition of the 10RM set is comparable to the final repetition of the other repetition ranges. This was in contrast to what was found in a similar study on bench press in which higher $V_{max1}$ was found at 10RM compared to 1 and 3RM (*Larsen, Haugen & van den Tillaar, 2022*). This difference could be due to the fact that bench press is an upper body exercise with different demands than in a lower body exercise like squats as it seems that in all ranges (1–10RM) the last repetition was maximal, as the barbell velocity in squats at each event was similar to the 1RM attempts.

Although the barbell velocity was similar in the last repetition in all ranges, a greater trunk lean was observed at 10RM at $V_0$ and $V_{max1}$ compared with 1RM. This could be a result of technical breakdown (*Halperin, Chapman & Behm, 2015*), due to the accumulated fatigue in the stabilising muscles of the spine resulting in inability to remain posture. This is logical, as the squat induces compressive force on the spine (*Schoenfeld, 2010*), while a high set duration may challenge cardio-respiratory fitness and the ability to maintain intra-abdominal pressure, which is important for maintaining a neutral spine (*Kibler, Press & Sciascia, 2006*). The current study only measured EMG of the erector spinae iliocostalis, and no differences were observed in amplitude between the different ranges (Table 3), but it was still possible that fatigue in other parts of the stabilising muscles may result in an increased trunk lean.

Differences in EMG amplitude were found only for the vastus lateralis, gastrocnemius, and soleus between 10RM and 1RM, in which at 10RM the EMG amplitude was lower than at 1RM. As shown by the similar barbell velocity between the two, it was to be expected that EMG amplitude would also be similar. In both ranges, the athlete cannot lift one repetition more due to fatigue. To complete high-load, short-duration lifts such as a 1RM squat, high-threshold motor units are necessitated (*Gundermann, 2012*) in accordance with Henneman's size principle (*Henneman, 1957*). It has been evidenced that EMG amplitude increases with increased loads in the squat (*Yavuz & Erdag, 2017*; *Paoli, Marcolin & Petrone, 2009*) and when comparing high *vs* low load (% relative to 1RM) sets (*Morton et al., 2019*; *Haun et al., 2017*), suggesting full motor unit recruitment to complete

the lift in the 1RM squat, which is not required in the first repetitions at 10RM. The time under tension significantly differs between 1 and 10RM sets, with 10RM sets lasting >15 s, compared to <5 s for 1RM sets (*Haff & Triplett, 2015*; *Shimano et al., 2006*). The extended time under tension during 10RM sets may increase metabolic demands due to the involvement of several large muscle groups in squats (*Kraemer & Ratamess, 2004*), as well as lead to muscle fatigue. This fatigue may cause failure of higher threshold motor units caused by acidosis and a reduced sensitivity of myofibrils to $Ca^{2+}$. Consequently, there is an increased reliance on lower threshold motor units to maintain force production and complete the last repetition (*Tesch et al., 1983*).

High load strength training (one to six repetitions) is heavily dependent on anaerobic energy release of the ATP-PC system, whereas training with more moderate loads (six to 12 repetitions) is mostly dependent on energy from ATP-PC and glycolysis, with some support from aerobic metabolism (*Kraemer & Ratamess, 2004*). With increased repetitions and muscular endurance requirement, the contribution from aerobic metabolism increases. Local muscle endurance training (*e.g.*, high number of repetitions) involves a higher contribution of energy from aerobic metabolism (*Kraemer & Ratamess, 2004*). Thus, during longer time under tension (*e.g.*, 10RM squat), a different support from other energy systems might be required, which can produce ATP again that contract the muscles in comparison to maximal lifts that are short in duration. Thus, motor unit recruitment and EMG amplitude in the 10RM squat might be influenced by locally fatiguing factors such as muscular ATP depletion and reduced oxidation due to the set duration (*Bloomer et al., 2006*; *Gorostiaga et al., 2012*).

The plantar flexors and knee extensors might be more suspectable to ATP depletion, as the squat starts with the knee extension and ankle plantar flexion movement (*van den Tillaar, Andersen & Saeterbakken, 2014*), prior to peak hip flexion. Alternatively, the initial repetitions of the 10RM set could be performed with a more "knee-dominant" technique (which also results in increased ankle dorsal flexion) (*Bryanton et al., 2012*), when perception of effort is low (*Halperin, Chapman & Behm, 2015*), increasing ATP depletion in the quadriceps and plantar flexors. The assumption is asserted from earlier research, which observed that when performing high-repetition squats, knee flexion decreases, while hip flexion increases as failure approaches (*Hooper et al., 2014*). When a muscle fatigues and becomes unable to exert required force, the athlete will self-organise to reduce the requirements of the fatigued muscle, thus increasing requirements of other muscles (*Hooper et al., 2014*; *Trafimow et al., 1993*). Based upon the differences observed in EMG and trunk lean, it could be speculated that different mechanisms lead to fatigue at 1 *vs* 10RM and when comparing a squat *vs* a bench press, thus affecting technical alterations as failure approaches.

Although fatigue-induced alterations occur in the squat (*Brice et al., 2020*; *Hooper et al., 2014*; *Trafimow et al., 1993*), the magnitude to which fatigue results in technical alterations is not possible to determine based on the current data, as measurement of maximal voluntary concentric contractions was not included. If fatigue is more local in the vastus lateralis and plantar flexors when performing 10RM *vs* 1 and 3RM, it could result in intra-set alteration in technique (*Larsen, Haugen & van den Tillaar, 2022*), as a

compensatory strategy to complete the set, such as reductions in peak knee flexion, while increasing peak hip flexion. Such alterations will change the moment arm for the hip and knee musculature, shifting the emphasis on hip extension moment during the sticking region. Unfortunately, the current study did not investigate the repetitions leading up to the last repetition, which should be done in future studies. However, strong caution should be exercised toward these interpretations with considerations for the inherent limitations of applying acute EMG data, especially to predict longitudinal outcomes (*Vigotsky et al., 2018*).

The current study has limitations that must be addressed. Firstly, the different repetitions of the sets were not analysed to investigate if EMG and joint kinematics change throughout the set, which would provide information on non-failure training (*Schoenfeld et al., 2021*). Secondly, although the subjects performed sets at different repetition ranges, their own body mass above the knee was not accounted for, which also influences absolute weight lifted in the squat. Thirdly, the use of measurements of local skeletal muscle oxygen saturation could aid in explaining the differences between 1 and 10RM due to possible different energy processes involved between the ranges.

## CONCLUSIONS

Based upon trunk lean and EMG amplitude of the vastus lateralis and plantar flexors in the last repetition, the last repetition of the 10RM squat seems different compared to 1RM, whereby local fatigue in the knee extensors and plantar flexors might lead to a different movement strategy. As such, based upon the principle of specificity, 10RM might induce different requirements during the different phases when ascending to complete the final repetition, in comparison to the sets performed with heavier loads (<3RM). This conclusion is caveated by sets being performed to volitional concentric failure.

### Funding
The authors received no funding for this work.

### Competing Interests
The authors declare that they have no competing interests.

### Author Contributions

- Hallvard Nygaard Falch conceived and designed the experiments, performed the experiments, analyzed the data, prepared figures and/or tables, authored or reviewed drafts of the article, and approved the final draft.
- Andreas Hegdahl Gundersen conceived and designed the experiments, performed the experiments, authored or reviewed drafts of the article, and approved the final draft.
- Stian Larsen conceived and designed the experiments, analyzed the data, authored or reviewed drafts of the article, and approved the final draft.
- Markus Estifanos Haugen conceived and designed the experiments, performed the experiments, authored or reviewed drafts of the article, and approved the final draft.
- Roland van den Tillaar conceived and designed the experiments, analyzed the data, prepared figures and/or tables, authored or reviewed drafts of the article, and approved the final draft.

## Human Ethics

The following information was supplied relating to ethical approvals (*i.e.*, approving body and any reference numbers):

Nord University granted ethical approval to carry out the study, along with approval from the Norwegian Centre of Research data.

## Data Availability

The raw data is available in the Supplemental File.

## Supplemental Information

Supplemental information for this article can be found online at http://dx.doi.org/10.7717/peerj.16865#supplemental-information.

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
