# Peer review of "Comparison of kinematics and electromyography in the last repetition during different maximum repetition sets in the barbell back squat"

_PeerJ, doi:10.7717/peerj.16865_

## Round 0.1 · original submission · Major Revisions

Dear Authors,

You will notice that we had three reviewers assess this work and I believe the extensive comments can help improve the quality of this manuscript.

While the reviewers believed the manuscript was interesting and the study relevant, you will find a general consensus that more details are required. As a result, this work requires a "major revision" prior to being considered for publication. In particular, the authors should consider the comments around EMG signal processing choices and overall amplitude interpretation. Reviewer comments can be found attached and listed below. We look forward to your reply and consideration of this work for publication upon further review.

Thank you,
Mike

·

Basic reporting

The general reporting is a good and good level of English is used throughout. The authors should be commended on making what could be quite a complex study design very easy to understand with good presentation of data.

References are sufficient, maybe a few instances were references could be included.

Very professional structure which greatly helps the readers interpretation of the results.

Experimental design

The work highlights a clear gap in the literature linking to previous work performed in the bench press but not in the squat. Therefore, provides clear aims and research questions which align with the amis and scope of the journal.

A rigorous investigation has been performed, some more discussion around the limitations would be beneficial, as highlighted below in the additional/specific comments. Further enhancing this work could then specifically highlight these as future research avenues.

The methods are well described and match included figures appropriately, I have added some specific comments below.

Validity of the findings

The finding are novel, achieving the aim and filling the gap highlighted they do have limited impact but it is still noteworthy information.

The study can be replicated by the methods described and really built upon.

All data has been provided is statistically sound and meaningful data which allows for robust interpretation by the readers. Further analysis could be performed to allow for more effective comparisons between maximal loads/set schemes.

Based on some of the limitations, the conclusions do fit well but more could be made of the results making the information more informative to readers and practitioners based off the results.

Additional comments

Abstract

L49 - Could you be specific around sports and daily living?

L50-51 – What technical requirements might change?

L54 – One thing which is not entirely clear in the early part of the study is that it is only concentric, this should be made clear here and in the introduction.

L67 – Could you add specific application of the results here?

Introduction

L71 – “general public” – could you be specific e.g. health related (long term sick/chronic illness) or older (geriatric/frail) and how strength or resistance training is crucial.

L74-76 – Could you note potential differences in the back squat? Bar position, squat depth, stance width

L79 – I know you have tried to write one as it is the first word, could you make it more coherent and change to “The 1RM…”

L29-80 – Could you also explore the thought of strength as a technical skill, specifically with reference to back squat.

L98 – Could you provide a reference for low vs high load training.

L98-105 – I think the below reference might be useful here.

Nuzzo, J.L., Pinto, M.D., Nosaka, K. et al. Maximal Number of Repetitions at Percentages of the One Repetition Maximum: A Meta-Regression and Moderator Analysis of Sex, Age, Training Status, and Exercise. Sports Med (2023). https://doi.org/10.1007/s40279-023-01937-7

L110 – What are the compensatory mechanisms you highlight?

L113 – What about the role decent and why this hasn’t been explored?

L128 – “personal preference” would it not be closer related to the training goal and hence the repetition continuum.

L130 – A study is inanimate, and it cannot aim, please amend accordingly.

L132 – Could you use the results by Larsen to inform a-priori sample size justification.


Methods

L154 – Power analysis would be beneficial and is likely low.

L155 – Spelling “strengPth”. Please amend.

L172 – Did all wear lifting shoes? How could this have affected the results?

L207-208 – What was the moving average window duration/length used for the root mean square calculation? Was this considered when looking at the frequency and resolution of the encoder for synchronisation?

L231 – The partial eta squared is good, but could you run pairwise effect sizes (Hedge’s g would be most appropriate) to identify the magnitude of the differences between each rep scheme.

Results

L237-238 – Don’t repeat results in text and tables. Additionally, be consistent with the number of decimal places.

Discussion

L278 – Change extremities to body

L322 & L335 – In my opinion to improve readability starting the paragraphs without “Furthermore” or “Moreover” would be beneficial. E.g. “High load strength training..” & “The plantar flexors…”

L328 – With the control of tempo (<2 seconds at the top), could there be a slight occlusion effect with the higher reps, how could this impact the fatigue observed.

L364 – There are probably too many limitations listed, almost makes the study not worthwhile, could you either reduce the number or explore why they were necessary in more detail.

L366 – Could you reference the Schoenfeld review

Schoenfeld BJ, Grgic J, Van Every DW, Plotkin DL. Loading Recommendations for Muscle Strength, Hypertrophy, and Local Endurance: A Re-Examination of the Repetition Continuum. Sports (Basel). 2021 Feb 22;9(2):32. doi: 10.3390/sports9020032. PMID: 33671664; PMCID: PMC7927075.

L369 – Nice point around system vs bar mass, does that matter for powerlifting?

L364 – Could you explore the fact that you have not observed the eccentric portion of the lift.

L376 – Specifically state, you can not infer training responses based on EMG.

Conclusions –

L380 – State explicitly that it is only observing the difference in the last repetition.

L385 – Concentric failure?

Table 2 – make the number of decimal places consistent (1dp/2dp).

Figure 1 – Could you add a diagram at the bottom of figure 1 to make it clear that it starts in the bottom position of the squat therefore only looking at the concentric phase.

Reviewer 2 ·

Basic reporting

General comments: The authors investigated the impact of loading on muscle activity and kinematics during the back squat. The study does add to the current literature and is seemingly well carried out, but does require some revisions before publication.

Introduction:
• Lines 73-79 get into a discussion with regards to powerlifting, however this is a very small niche and while it is the highest level with which 1RM strength is displayed, I think it is a weak argument as to why this study is important. I think the authors have a strong argument as to the study, but this is not quite the right angle. Instead, the papers the authors already cite articles that discuss resistance training, and likely make mention of using a % of 1RM to program lifts. I think a better argument for the authors to make in the introduction is that while % of 1RM is often used to prescribe training loads, we do not know that technique remains consistent across the loading spectrum. This can then be extended to using different RMs as a programming methodology needing explored.
o These two papers may be of some assistance in forming that argument:
o https://www.tandfonline.com/doi/abs/10.1080/17461391.2022.2081093
o https://link.springer.com/article/10.1007/s11332-021-00764-5
• The authors mention barbell velocity in line 135, but the idea is not fully developed. I think a more thorough explanation of the study cited here is warranted (see comment below).
• The authors should add a clear hypothesis statement of their own. I believe the best way to do this is to move lines 132-135 up to the previous paragraph and then only discuss their own prediction in the final paragraph.
• Another idea to consider here that would make this a stronger paper is a short discussion regarding fatigue vs. non-fatigue training, as it dominates your discussion.
• To sum up my comments, I think the authors should consider re-working this introduction centered around 3 topics: change in emg with load, barbell velocity changes with load, and fatigue and then end with a strong hypothesis statement. This would create a much stronger paper.

Experimental design

Materials and Methods:
• Lines 141-146 are a run-on sentence and should be split into two sentences.
• Line 155 typo
• Lines 154 – 184 please separate this into the two testing sessions and provide a greater level of detail with regard to how loading was determined for each lifter. It is difficult to understand the way the sessions were conducted. Did the lifters complete the same loads on both days? Were 1, 3, 6, and 10 RM estimated or calculated in some manner. If calculated how?
• Why was the emg data not normalized? This is a big sticking point as it goes against the vast majority of EMG recommendations regarding the use of the technology. If the authors have a specific reason for not normalizing, it should be presented in the paper and cited.
o https://www.intechopen.com/chapters/40113
o https://www.sciencedirect.com/science/article/pii/S0161475499700321

Validity of the findings

Discussion:
• Interesting discussion overall and I think you draw reasonable conclusions based on the data collected. Please make the revisions to the introduction and I believe that will create a much stronger paper overall given the current discussion.
• Line 314 - 320 are unclear, please revise these for clarity. I believe the authors are trying to make that point that during a 10RM set the total TUT is greater and there may be a greater metabolic demand as a result, but that is hard to get from the current wording.

Additional comments

General comments: The authors investigated the impact of loading on muscle activity and kinematics during the back squat. The study does add to the current literature and is seemingly well carried out, but does require some revisions before publication.

Reviewer 3 ·

Basic reporting

In general, the language used in the study is complex and difficult to understand.
In the introduction section, the purpose of the study was explained but no information about the hypotheses was provided.
Although the introduction is suitable for explaining the purpose of the study, it should be more clearly explained what the difference is from Larsen and Haugen's study and why such a study is needed. Otherwise it would be possible but pointless to repeat the same study for all type of exercises.

Experimental design

the rest periods between sets should be add since it is very important to recover from fatigue.

Validity of the findings

Although the study requires serious effort and yields important data, the data is difficult to understand and interpret.
Especially the data in table 3 seems difficult to understand and interpret. In the description of the table, it is stated what the '*' sign stands for, but not what the 'T' sign stands for. There are findings which were not discussed in the discussion.
Using figures instead of tables could be easier to understand and interpret.
In the second paragraph of the discussion, when comparing with a previous study, there is an implication that the data taken in the other study does not reflect the data actually intended to be taken (290-291).
This may not be an appropriate way to explain the different results obtained in different studies.
As the conclusion of the study it was emphasized that 10RM might locally fatigue vastus lateralis and plantar flexors, explaining the lower EMG amplitude. Could the possible explanation for lower emg amplitudes simply be lower loads used? or any other reason you already discussed in detail in the discussion.

Additional comments

Thanks for your effort and contributions to the topic.

---

## Round 0.2 · Minor Revisions

Dear Dr. Falch,

You will be pleased to find that all reviewers were in agreement and commend your team on a much improved version of the manuscript. I am in agreement with reviewer #2 and would just like to see a short comment added to the manuscript about the EMG normalization process. Please consider a statement in the methods, and/or further comment in the discussion about this point. After which, I believe the work will be immediately accepted.

Thank you kindly,
Mike

·

Basic reporting

Much improved. Happy with the changes that have been made.

Experimental design

Much improved. Happy with the changes that have been made.

Validity of the findings

Much improved. Happy with the changes that have been made.

Additional comments

Much improved. Happy with the changes that have been made.

Reviewer 2 ·

Basic reporting

The authors have made quality revisions based on my and the other reviewers' comments which has created a very good introduction. I have no further comments for this section.

Experimental design

With regards to the EMG normalization question, I agree it is context dependent and I agree that the authors do not need to normalize the data for this to be publishable. Just please in one or two sentences add that to the methodology to explain it. I personally like that it challenges the status quo of EMG protocols, so please explain why because your argument is sound.

Validity of the findings

No further comments, thank you for the clarification on the TUT.

Additional comments

Well done with your revisions.

Reviewer 3 ·

Basic reporting

the amendments made in line with the recommendations were deemed appropriate.

Experimental design

the amendments made in line with the recommendations were deemed appropriate.

Validity of the findings

the amendments made in line with the recommendations were deemed appropriate.

Additional comments

the amendments made in line with the recommendations were deemed appropriate.

---

## Round 0.3 · accepted · Accept

Thank you for considering our requests and congratulations on your work!